# Quantitative Evaluations of Hydrogen Diffusivity in V-X (X = Cr, Al, Pd) Alloy Membranes Based on Hydrogen Chemical Potential

**DOI:** 10.3390/membranes11010067

**Published:** 2021-01-18

**Authors:** Asuka Suzuki, Hiroshi Yukawa

**Affiliations:** 1Department of Materials Process Engineering, Graduate School of Engineering, Nagoya University, Furo-cho, Chikusa-ku, Nagoya 464-8603, Japan; 2Department of Materials Design Innovation Engineering, Graduate School of Engineering, Nagoya University, Furo-cho, Chikusa-ku, Nagoya 464-8603, Japan; hiroshi@nagoya-u.jp

**Keywords:** hydrogen permeation, vanadium, mobility, chemical potential, first principle calculation

## Abstract

Vanadium (V) has higher hydrogen permeability than Pd-based alloy membranes but exhibits poor resistance to hydrogen-induced embrittlement. The alloy elements are added to reduce hydrogen solubility and prevent hydrogen-induced embrittlement. To enhance hydrogen permeability, the alloy elements which improve hydrogen diffusivity in V are more suitable. In the present study, hydrogen diffusivity in V-Cr, V-Al, and V-Pd alloy membranes was investigated in view of the hydrogen chemical potential and compared with the previously reported results of V-Fe alloy membranes. The additions of Cr and Fe to V improved the mobility of hydrogen atoms. In contrast, those of Al and Pd decreased hydrogen diffusivity. The first principle calculations revealed that the hydrogen atoms cannot occupy the first-nearest neighbor T sites (T1 sites) of Al and Pd in the V crystal lattice. These blocking effects will be a dominant contributor to decreasing hydrogen diffusivity by the additions of Al and Pd. For V-based alloy membranes, Fe and Cr are more suitable alloy elements compared with Al and Pd in view of hydrogen diffusivity.

## 1. Introduction

Hydrogen energy needs to be utilized effectively to realize a decarbonized society [1]. Since hydrogen does not exist as a simple substance in nature, it needs to be manufactured industrially. Even when hydrogen gas is manufactured or generated through any routes, it is necessary to purify the product gas for usage in the fuel cells since it always contains not only hydrogen gas but also by-product gases [2]. Hydrogen gas with high purity is obtained by separation and purification through hydrogen-permeable dense metallic membranes [3]. Pd-based alloy membranes are the most representative metallic membranes used for the purification of hydrogen gas [4,5].

Recently, group 5 metal-based alloy membranes have been developed to avoid using high-cost precious metal Pd. The hydrogen permeability of group 5 metals with a body-centered cubic (BCC) crystal structure is much more superior to Pd and Pd-based alloy membranes [6]. However, group 5 metals have high hydrogen solubility and become significantly brittle under hydrogen atmosphere (hydrogen-induced embrittlement) [7,8]. The presence of the ductile-to-brittle transition hydrogen concentration (DBTC) has been found by small punch testing under hydrogen atmospheres at elevated temperatures [8,9]. The small punch absorption energy of group 5 metals decreases drastically at around 0.2 (H/M). Hydrogen solubility needs to be optimized so that the hydrogen concentration does not exceed 0.2 (H/M) under operating pressure and temperature conditions of metal membranes. The elements with low affinity for hydrogen are usually added to control hydrogen solubility.

Considering the influence of the alloy elements on the microstructure and hydrogen diffusivity, single-phase solid solution alloys are considered to be preferable. To form the solid solution alloys and control hydrogen solubility, alloy elements with large solid solubility limits in base metals are preferable. Among group 5 metals, V forms solid solutions with many elements in wide ranges of alloy compositions. Therefore, various V-based solid solution alloys such as V-Ni [10,11,12], V-Cr [12], V-Al [12,13], V-Pd [14], V-Fe [15], V-W [16], and V-W-Mo [17] alloys have been investigated. It is more suitable that the alloy elements enhance hydrogen diffusivity in V. It is reported that the addition of Fe to V enhances hydrogen diffusivity below approximately 700 K [15].

In the present study, the hydrogen solubility and permeability of V-Al, V-Cr, and V-Pd alloy membranes were investigated systematically to examine the effects of alloy elements on hydrogen diffusivity quantitatively. The mobility of hydrogen atoms, *B*, in these alloy membranes was evaluated by the following equation [18]:(1)$$J=\frac{RTB}{2L}\int_{c_{2}}^{c_{1}}c\frac{d\ln\left(P/P^{0}\right)}{dc}dc=\frac{RTB}{2L}f_{\mathrm{PCT}}$$

This equation is based on the diffusion equation based on the chemical potential of hydrogen [18]. Here, *J* is hydrogen flux through the membrane, *L* is the membrane thickness, *R* is the gas constant, *T* is absolute temperature, *c* is hydrogen concentration, *c*_1_ and *c*_2_ are hydrogen concentrations at feed and permeation sides of the membrane, *P* is hydrogen pressure, *P*^0^ is the standard hydrogen pressure (101325 Pa), and *f*_PCT_ represents the integral term in Equation (1).

The first principle calculations were also carried out to investigate the interactions between hydrogen atoms and alloy elements in a BCC V crystal lattice.

## 2. Materials and Methods

### 2.1. Experimental Procedure

V-4 mol%Cr, V-14 mol%Cr, V-23 mol%Cr, V-5.5 mol%Al, V-16 mol%Al, V-20 mol%Al, V-25 mol%Al, V-10 mol%Pd, and V-15 mol%Pd alloy ingots were melted under an argon gas atmosphere using a tri-arc furnace. The purity of starting materials is 99.9 mass% for vanadium (Taiyo Koko Co., Ltd., Tokyo, Japan), 99.88 mass% for chromium (Tosoh Co., Ltd., Tokyo, Japan), 99.999 mass% for aluminum, and 99.95 mass% for palladium (Tanaka Kikinzoku Kogyo K.K., Tokyo, Japan). The chemical compositions of each sample analyzed by a scanning electron microscopy/energy dispersion X-ray spectroscopy are summarized in Table 1. In the equilibrium phase diagrams for V-Cr, V-Al, and V-Pd binary systems [19,20,21], these alloys consist of a single phase with a BCC crystal structure. Figure 1 shows the X-ray diffraction (XRD) profiles of these alloys. Only peaks derived from the BCC crystal structure were detected.

The PCT curves of the alloys were measured using a Sieverts-type apparatus to examine hydrogen solubility and to estimate the PCT factor (*f*_PCT_). A small bulk cut from the sample ingot was set into a sample cell, and then the sample cell was set on the apparatus and evacuated. The activation process for the sample was carried out before measuring the PCT curves. The detailed activation process applied in this study was given elsewhere [15]. The PCT measurements were carried out at 473~673 K.

The samples for hydrogen permeation tests were fabricated by cutting the as-cast ingots of V-4 mol%Cr, V-14 mol%Cr, V-23 mol%Cr, V-5.5 mol%Al, V-16 mol%Al, V-20 mol%Al, V-25 mol%Al, V-10 mol%Pd, and V-14.5 mol%Pd alloys into disks with approximately 12 mm in diameter using a wire-cut electrical discharge machine (EDM). The damaged surface layer of the disk samples caused by EDM was removed by mechanical polishing with alumina abrasive papers and heat treatment in a vacuum at 1273 K for 24 h. The surfaces of disk samples were treated by mechanical polishing using alumina abrasive papers and buffing with diamond slurry (9 μm and 1 μm). The final thicknesses (*L*) of each sample were 0.500 mm for V-4 mol%Cr, 0.428 mm for V-14 mol%Cr, 0.521 mm for V-23 mol%Cr, 0.550 mm for V-5.5 mol%Al, 0.562 mm for V-16 mol%Al, 0.425 mm for V-20 mol%Al, 0.504 mm for V-25 mol%Al, 0.521 mm for V-10 mol%Pd, and 0.366 mm for V-14.5 mol%Pd, respectively. A radio-frequency (RF) magnetron sputtering apparatus was used for coating the Pd-27mol%Ag alloy on both sides of the membrane samples. The sputtering was carried out for 180 s at 573 K under Ar gas atmosphere of 1 Pa so that the thickness of the overlayer was approximately 200 nm. The reason why the Pd-Ag alloy was selected for the overlayer instead of pure Pd was given in the previous study [15].

Hydrogen permeability was measured by the gas permeation method applying differential pressures. The G1 grade hydrogen gas was used for the tests. The testing conditions including the temperature and the hydrogen pressures are summarized in Table 2, Table 3 and Table 4. Detailed methods of the hydrogen permeation test are described in the previous study [22]. To eliminate the effect of the difference in the membrane thickness, the normalized hydrogen flux (*J*·*L*) was evaluated in this study. Note that the unit of the normalized hydrogen flux in this study is mol H m^−1^ s^−1^ instead of mol H_2_ m^−1^ s^−1^.

### 2.2. First Principle Calculation

Interactions between alloy elements and hydrogen atoms in vanadium (BCC) crystal lattice were investigated by the first principle calculations based on the density functional theory (DFT). Supercell models were constructed by 2 × 2 × 2 of the BCC crystal structure of vanadium. The atom at the center of the supercell was substituted for alloy elements (X = Cr, Pd, Al, and Fe). This model is denoted as V_15_X_1_ assuming V-X solid solution alloy. One hydrogen atom was put into the first-, second-, and third-nearest neighbor tetrahedral interstitial sites (T1, T2, and T3 sites) of the X atom. These models are denoted as V_15_X_1_H_1_(T1), V_15_X_1_H_1_(T2), and V_15_X_1_H_1_(T3) and are treated as hydrogen solid solution phases with the hydrogen concentration of 0.0625 (H/M). The detailed model explanation is given in the previous report [23].

The CASTEP code was used for geometry optimizations and energy calculations for all these models [24]. A plane-wave pseudopotential method implemented in the CASTEP code is applied. The pseudopotential used in this study was the Vanderbilt ultra-soft type. The Perdew–Burke–Ernzerhof generalized gradient approximation (GGA-PBE) was applied. The cutoff energy and k-point grid were set at 400 eV and 8 × 8 × 8, respectively.

Firstly, V_15_X_1_ models were fully optimized, and the lattice parameters of V_15_X_1_ models were calculated. The lattice parameters of the V_15_X_1_H_1_(T1) model were estimated using the lattice parameters of V_15_X_1_ models and considering the volume expansion induced by the occupation of hydrogen atoms into the BCC crystal lattice (approximately 2.6 Å^3^ per 1 (H/M)) [25]. The optimized V_15_X_1_ models were isotopically expanded, and a hydrogen atom was inserted into the T1 site. Then, the constructed V_15_X_1_H_1_(T1) models were geometrically optimized. Only the positions of the atoms inside the supercell were optimized while the lattice parameters were constrained.

The changes in the total energy induced by hydrogen insertion into T1, T2, and T3 sites were calculated with the following equation:(2)$$\Delta E=E\left(V_{15}X_{1}H_{1}\right)-E\left(V_{15}X_{1}\right)-\frac{1}{2}E\left(H_{2}\right)$$
where *E*(V_15_X_1_H_1_), *E*(V_15_H_1_), and *E*(H_2_) are the total energies of V_15_X_1_H_1_, V_15_H_1_, and H_2_ models, respectively.

## 3. Results

### 3.1. Experimental Results

Figure 1 shows the X-ray diffraction (XRD) profiles of these alloys. Only peaks derived from the BCC crystal structure were detected, suggesting that all the samples are composed of single phases with BCC crystal structures.

Figure 2 shows the pressure-composition isotherms (PCT curves) of (a) V-4 mol%Cr, (b) V-14 mol%Cr, and (c) V-23 mol%Cr alloys. The V-Cr alloys exhibited PCT curves with inverse sigmoid shapes. There were clear inflection points even when Cr concentration was 23 mol%. Hydrogen solubility decreased with increasing temperature, indicating that dissolutions of hydrogen into V-Cr alloys were exothermic reactions.

Figure 3 presents the PCT curves of (a) V-5.5 mol%Al, (b) V-16 mol%Al, (c) V-20 mol%Al, and (d) V-25 mol%Al alloys. The V-5.5 mol%Al alloy also exhibited typical PCT curves with inverse sigmoid shapes. However, in the PCT curves of V-16 mol%Al, V-20 mol%Al, and V-25 mol%Al alloys, clear inflection points were not observed. For example, although the PCT curve for V-20 mol% Al at 573 K was an upward convex curve at low pressure and a downward convex curve at high pressure, its boundary was not clear. These alloys had a steeper slope around the inflection points compared with V-Cr alloys. As the temperature increased, the PCT curves shifted toward the upper left region. Thus, the dissolution of hydrogen into V-Al alloys was also exothermic reactions.

Figure 4 shows the PCT curves of (a) V-10 mol%Pd and (b) V-14.5 mol%Pd alloys. The PCT curves of these alloys were upward convex curves under the whole hydrogen pressure region tested in this study (approximately up to 2 MPa). The temperature dependence of the PCT curves for these alloys also indicated that exothermic reactions took place upon hydrogen uptake into them.

The pressure of gaseous hydrogen in equilibrium with hydrogen atoms in metals and alloys with the hydrogen concentration of 0.2 (H/M), i.e., DBTC (*P* (0.2)), were quantified from the PCT curves shown in Figure 2, Figure 3 and Figure 4. Figure 5 shows the change in *P* (0.2) for (a) V-Cr, (b) V-Al, and (d) V-Pd alloys with reciprocal temperature. For comparison, the results for pure V [15] are also shown in the figure. The addition of Cr, Al, and Pd increased *P* (0.2). The logarithmic hydrogen pressure decreased linearly as the inverse of temperature increased. The linear correlations in Figure 5 are known as the van’t Hoff equation in Equation (3).
(3)$$\ln\frac{P\left(c\right)}{P^{0}}=\frac{2\Delta \bar{H}\left(c\right)}{RT}-\frac{2\Delta \bar{S}\left(c\right)}{R}$$
where *P*(*c*) is the pressure of gaseous hydrogen in equilibrium with hydrogen atoms in metals and alloys with the hydrogen concentration of *c*. $\Delta \bar{H}\left(c\right)$ and $\Delta \bar{S}\left(c\right)$ are the partial molar enthalpy and entropy changes of hydrogen for forming hydrogen solid solution phase with hydrogen concentration of *c*. From the regression lines in Figure 5, $\Delta \bar{H}\left(0.2\right)$ and $\Delta \bar{S}\left(0.2\right)$, which are denoted as $\Delta \bar{H}$_0.2_ and $\Delta \bar{S}$_0.2_ in the present study, were quantified.

Figure 6 presents the changes in (a) $\Delta \bar{H}$_0.2_ and (b) $\Delta \bar{S}$_0.2_ for V-Cr, V-Al, and V-Pd alloys with mole fractions of each alloy element. For comparison, the values for pure V and V-Fe alloys are also shown in the figure [15]. $\Delta \bar{H}$_0.2_ increased as the mole fractions of each alloy element increased (Figure 6a) because Fe, Cr, Al, and Pd exhibit lower affinity to hydrogen than V. The additions of Fe and Pd increased the partial molar enthalpy change more largely than those of Cr and Al. $\Delta \bar{H}$_0.2_ increased almost linearly with the mole fractions of Fe and Cr, but it increased along downward convex curves with increasing Al and Pd mole fractions. When the amount of alloy elements was small, V-Al alloy showed the lowest $\Delta \bar{H}$_0.2_ among those elements. However, the value for the V-Al alloy became higher that than for the V-Cr alloy when the mole fraction was over approximately 20 mol%.

In contrast, $\Delta \bar{S}$_0.2_ was almost constant at approximately −47 J mol H^−1^ K^−1^ (−94 J mol H_2_^−1^ K^−1^) and independent of types and additive amount of alloy elements. The absolute value of the partial molar entropy was close to the entropy of hydrogen gas in the standard state (~130 J mol H_2_^−1^ K^−1^) [26], indicating that the partial molar entropy change is mainly attributable to the loss of hydrogen gas with high entropy. Thus, the alloying effects on hydrogen solubility can be understood by $\Delta \bar{H}$_0.2_.

The PCT curves in Figure 2, Figure 3 and Figure 4 were used to quantify the PCT factors (*f*_PCT_) in Equation (1) under the conditions listed in Table 2. A detailed explanation of the way to quantify *f*_PCT_ is given elsewhere [27]. Figure 7, Figure 8 and Figure 9 present the relationship between normalized hydrogen flux (*J·L*) and *f*_PCT_ for V-Cr, V-Al, and V-Pd alloys. As shown in these figures, there are almost linear correlations between *J·L* and *f*_PCT_. The slope of the regression lines increased with the increasing temperature and mole fraction of alloy elements. It is noted that the addition of Al to V obviously reduced the slope of the regression lines, which is a different trend from V-Cr alloy membranes.

The mobility of hydrogen atoms (*B*) was estimated from the slopes of the regression lines in Figure 7, Figure 8 and Figure 9. *B* in each alloy at each temperature is summarized in Table 5. For comparison, *B* in pure Pd and Pd-23 Ag alloy [28] is also shown in the Table. Figure 10 presents an Arrhenius plot of *B* in (a) V-Cr, (b) V-Al, and (c) V-Pd alloy membranes. For comparison, the hydrogen mobility in pure V is also shown in the figure [15]. The logarithmic mobility decreased almost linearly with increasing reciprocal temperature, indicating that the temperature dependence on *B* was described by the Arrhenius equation as indicated in Equation (4).
(4)$$B=B_{0}\exp\left(-\frac{Q}{RT}\right)$$
where *Q* is the activation energy required for hydrogen to diffuse, and *B*_0_ is the pre-exponential factor. The addition of Cr to V increased hydrogen mobility in the range of 573–673 K (Figure 10a). The mobility increased significantly especially at lower temperatures by adding Cr. The addition of 5.5 mol% Al to V also increased the mobility slightly at lower temperatures (Figure 10b). However, the mobility decreased by adding Al further. It is noted here that the regression lines of V-16 mol%Al, V-20 mol%Al, and V-25 mol%Al alloys were almost parallel. The addition of Pd to V decreased the mobility of hydrogen atoms in the range of 573–673 K (Figure 10a).

*Q* and *B*_0_ for each alloy were quantified based on Equation (4) and are summarized in Table 6. For comparison, *Q* and *B*_0_ for pure Pd and Pd-23 Ag [28] in similar temperature ranges are also shown in the Table. Figure 11 presents changes in (a) *Q* and (b) *B*_0_ with the mole fraction of each alloy element. For comparison, the results of pure V and V-Fe alloy membranes [15] are shown in the figure. *Q* decreased almost linearly with the mole fractions of Fe, Pd, and Cr (Figure 11a). Among these elements, the addition of Fe decreased *Q* the most sharply. The addition of Al to V exhibited an anomalous effect on *Q*. *Q* decreased almost linearly when the Al mole fraction was up to around 16 mol% but became almost constant when the Al mole fraction was over 16 mol%. In contrast, the logarithmic *B*_0_ decreased almost linearly with the mole fraction of all the alloy elements. The additions of Fe and Pd decreased *B*_0_ more sharply than those of Al and Cr.

Figure 12 presents the relationship between logarithmic *B*_0_ and linear *Q*. As the data points are plotted in the upper left region, hydrogen diffusivity is higher. There is a roughly positive correlation between logarithmic *B*_0_ and linear *Q*, and both *Q* and logarithmic *B*_0_ decreased by adding alloy elements (i.e., the plots move toward the lower left region in the figure). In particular, the results of pure V, V-Fe alloy, and V-Cr alloy are located almost on a single line (the broken line in the figure). The linear correlation between *Q* and logarithmic *B*_0_ is known as the Meyer–Neldel rule [29]. The plot for V-5.5Al is located on the line, but the results of V-Al alloy with higher Al mole fractions deviate from the line. The plots are located below the broken line and drop vertically with constant *Q* in the figure. In addition, the results for V-Pd alloys are also located below the broken line. These results indicate that the hydrogen diffusivity in V-Al and V-Pd alloys is lower than that in V-Fe and V-Cr alloys.

### 3.2. First Principle Calculations

Figure 13 shows the (a,b) V_15_Cr_1_H_1_(T1) and (c,d) V_15_Pd_1_H_1_(T1) models (a,c) before and (b,d) after geometry optimizations. In the case of the V_15_Cr_1_H_1_(T1) model, the V and Cr atoms around the hydrogen atoms were relaxed slightly, but the hydrogen atoms still occupied the T1 site of Cr. The hydrogen atom moved slightly to the position of Cr (Figure 13a,b), indicating attractive interactions operating between hydrogen and Cr atoms. These results are similar to the geometry optimized V_15_Fe_1_H_1_(T1) model reported in the literature [23]. In contrast, in the case of the V_15_Pd_1_H_1_(T1) model, it is interesting that the hydrogen atom in the V_15_Pd_1_H_1_(T1) model moved from the T1 site to the second-nearest neighbor T site (T2 site) of Pd. The Pd in BCC V exhibits a strong repulsive interaction to hydrogen atoms, although pure Pd with a face-centered cubic (FCC) crystal structure exhibits high affinity for hydrogen. Similar geometry optimizations were reported in the V_15_Al_1_H_1_(T1) model [23].

Figure 14 shows the total energy change (∆*E*) for the insertion of a hydrogen atom into the T1, T2, and T3 sites of alloy elements (Fe, Al, Cr, and Pd). Here, the hydrogen atoms in V_15_Al_1_H_1_ and V_15_Pd_1_H_1_ models were constrained not to move to the T2 site. Therefore, the total energy change of V_15_Al_1_H_1_ (T1) and V_15_Pd_1_H_1_ (T1) models (in parentheses) cannot be directly compared with those of other models. The total energy change of the V_16_H_1_ model was approximately −0.36 eV, which is in good agreement with the experimental partial molar enthalpy change of hydrogen for hydrogen dissolution (−0.32 eV H atom^−1^) [25]. Replacing the central V atom with Cr or Fe atom increased the total energy changes to approximately −0.25 eV and −0.30 eV. The total energy decreased slightly as the distance of hydrogen atoms from Fe or Cr atoms increased, but the energy level of the V_15_X_1_H_1_ (T3) model is still comparable to that of the V_15_X_1_H_1_ (T1) model. These results suggest that the additions of Fe or Cr to V increased the potential of hydrogen atoms in V uniformly, which would coincide with the linear increase in $\Delta \bar{H}$_0.2_ of V-Fe and V-Cr alloys (Figure 6). The total energy changes of V_15_Al_1_H_1_ (T1) and V_15_Pd_1_H_1_ (T1) models (denoted in parentheses) are around 0 eV and much higher than those of other models. It is noted here that the total energy changes of V_15_Al_1_H_1_ (T2 and T3) and V_15_Pd_1_H_1_ (T2 and T3) models are comparable to those of V_16_H_1_ and V_15_Cr_1_H_1_ models, respectively, indicating that the strong repulsive interactions between H and Al or Pd are localized only near these elements.

## 4. Discussion

The additions of Cr, Al, and Pd to V decreased the activation energy required for hydrogen to diffuse (*Q*) (Figure 11a). Similar results are also reported that the additions of alloy elements with lower affinity to hydrogen to group 5 metals decrease *Q*. For example, the additions of W, Mo, and Ru to Nb decrease the activation energy required for hydrogen to diffuse [30]. It is understood that the mitigation of diffusion barriers is caused by the increase in the potential of the hydrogen atoms at interstitial sites of the BCC Nb crystal lattice [31]. The additions of W, Mo, and Ru with lower affinity to hydrogen than Nb increase the potential of hydrogen atoms in Nb. The potential of the hydrogen atoms at interstitial sites can be understood by the equilibrium hydrogen pressure: higher hydrogen pressure indicates that hydrogen atoms are presented at shallow potential valleys in metals and alloys. The additions of Cr, Al, and Pd increased the equilibrium hydrogen pressure (Figure 5).

It was hypothesized that the activation energy required for hydrogen to diffuse decreases due to the increase in the potential of hydrogen in the BCC V crystal lattice. The increase in the equilibrium hydrogen pressure is attributable to that in the partial molar enthalpy change for forming hydrogen solid solution ($\Delta \bar{H}$_0.2_) (Figure 6). In order to verify this hypothesis quantitatively, the change in *Q* was plotted as a function of $\Delta \bar{H}$_0.2_ in Figure 15a. There is a trend that *Q* decreased with increasing $\Delta \bar{H}$_0.2_. The plots for most of the alloys, except for V-14.5 mol%Pd and V-25 mol% Al alloys, are aligned on a line (broken line in the figure). These results support the hypothesis that the increase in the potential of hydrogen in the V crystal lattice by adding alloy elements with lower affinity to hydrogen contributes dominantly to the decrease in *Q*, as illustrated schematically in Figure 15b. The plots for V-14.5 mol%Pd and V-25 mol% Al alloys are located above the broken line, indicating that another mechanism acts so as not to decrease the activation energy by the additions of Pd and Al to V.

It is reported that the addition of Al to V above 20 mol% decreased hydrogen permeability significantly [13]. Transmission electron microscopy (TEM) observations revealed the presence of some precipitates in the V-30 mol% Al alloy in crystal grains and at grain boundaries and sub-boundaries. However, the amount of those precipitates is too small to explain the significant reduction in the hydrogen permeability above 20 mol% Al, suggesting that the reason for the phenomenon lies in the matrix of V-Al solid solution [13]. In other words, the significant decrease in hydrogen diffusivity in V-Al and V-Pd alloys in the present study, as well as the significant reduction in hydrogen permeability of V-Al alloys in the previous study, may be caused by the interactions between alloy elements and hydrogen atoms in the BCC V crystal lattice.

The first principle calculations revealed that hydrogen atoms could not occupy the first-nearest neighbor T sites (T1 sites) of Al [23,32] and Pd (Figure 13) atoms in BCC V due to strong repulsive interactions. This strong repulsive interaction acted only when the hydrogen atoms occupied the T1 sites of Al and Pd in BCC V (Figure 14). These results are experimentally supported by the PCT curves of V-Al alloys (Figure 3): the slope of the PCT curves of V-16 mol%Al, V-20 mol%Al, and V-25 mol%Al was significantly sharp, and hydrogen concentrations in the high hydrogen pressure region were low compared with V-Cr alloys.

The T1 sites neighboring Al or Pd atoms are blocked by strong repulsive interactions with hydrogen. This blocking effect of Al or Pd atomsdecreases hydrogen diffusivity (Figure 10 and Figure 12). Figure 16 presents a schematic illustrations of (001) plane projection on a BCC crystal lattice. Arrows represent the jumps of hydrogen atoms between adjacent interstitial T sites. Hydrogen atoms cannot jump to the T sites near Pd and Al atoms represented by X symbols in Figure 16. Then, hydrogen atoms need to diffuse while bypassing around Pd or Al atoms (black arrows in Figure 16). As a result, diffusion distance becomes longer, and the apparent mobility of hydrogen atoms decreases.

Both Al and Pd in BCC V inhibit the diffusion paths of hydrogen atoms by blocking the T1 sites but have different effects on hydrogen diffusivity. The addition of a larger amount of Al to V does not decrease the activation energy required for hydrogen to diffuse while decreasing the pre-exponential factor (Figure 11 and Figure 12). The addition of Pd to V decreases both the activation energy and the pre-exponential factor (Figure 11), whereas the pre-exponential factor of V-Pd alloys decreases with the activation energy required for hydrogen to diffuse significantly compared with V-Fe and V-Cr alloys (Figure 12). One of the possible reasons for the difference is the short-range ordering of Al in BCC V. In the present study, all the V-Al alloys were composed of a single phase with the BCC crystal structures, as identified with XRD measurements (Figure 1). However, the presence of the ordered phase with A15 or B2 crystal structures in V-rich V-Al alloys below approximately 873 K was suggested in the previous studies [20]. Although such phases were not detected in the present study, Al in BCC V might be ordered in a short range. In such a case, hydrogen atoms could be present in almost the same potential valley even at high Al concentration. Then, the activation energies are almost constant regardless of the Al concentration. In contrast, no ordered phase has been suggested in the V-Pd binary system up to approximately 20 mol%Pd. Pd atoms can substitute V atoms randomly. Even though hydrogen atoms cannot occupy the T1 site of Pd in BCC V, the potential level of the hydrogen atoms in interstitial sites except for the T1 site is increased by the addition of Pd, resulting in a decrease in the activation energy. However, further investigations will be needed to demonstrate the presence or absence of the short-range ordering of alloy elements in BCC V and clarify the difference in the alloying effects of Al and Pd on hydrogen diffusivity in V.

Group 5 metals-based alloy membranes are designed so as to reduce hydrogen solubility for preventing the hydrogen-induced embrittlement while increasing the slope of the PCT curves for enhancing the hydrogen flux [33]. In order to control the PCT curves, the alloy elements with lower affinity to hydrogen than base metal need to be added. V-based solid solution alloy membranes such as V-Fe, V-Cr, V-Al, and V-Pd alloy membranes are widely investigated [12,13,14,15]. It is suitable that the alloy elements not only reduce hydrogen solubility but also enhance hydrogen diffusivity. The present study revealed that the alloying effects of Cr and Al on hydrogen solubility were smaller than those of Fe and Pd (Figure 6). In order to reduce hydrogen solubility, a large amount of Cr and Al needs to be added to V compared with Fe and Pd. However, a large amount of Al in V decreases hydrogen diffusivity (Figure 11 and Figure 12) by the blocking effects required for hydrogen to diffuse (Figure 15). Furthermore, the addition of Pd to V reduces hydrogen solubility as much as that of Fe but decreases hydrogen diffusivity (Figure 11 and Figure 12). Thus, Fe and Cr are more suitable alloy elements for V-based alloy membranes than Al and Pd in view of hydrogen diffusivity.

## 5. Conclusions

In the present study, alloying effects of Cr, Al, and Pd on hydrogen diffusivity in V were estimated quantitatively by quantifying hydrogen mobility and compared with the previously reported results of V-Fe alloy membranes. The additions of Cr and Fe to V enhance hydrogen diffusivity in the temperature range of 573~673 K, whereas those of Al and Pd decrease hydrogen diffusivity. From the correlation between the activation energy required for hydrogen to diffuse and the partial molar enthalpy change of hydrogen for forming hydrogen solid solution, the enhancement in hydrogen diffusivity by the addition of Cr is caused by the increase in the potential of hydrogen atoms in interstitial sites in the BCC V crystal lattice. The first principle calculation revealed that hydrogen atoms can occupy the first-nearest neighbor T site (T1 site) of Cr or Fe in the BCC V crystal lattice but cannot occupy the T1 site of Al or Pd. The blocking effects of Al and Pd inhibit hydrogen diffusion. In view of hydrogen diffusivity, Cr and Fe are more suitable alloy elements for V-based hydrogen-permeable alloy membranes than Al and Pd.

## Figures and Tables

**Figure 1 membranes-11-00067-f001:**
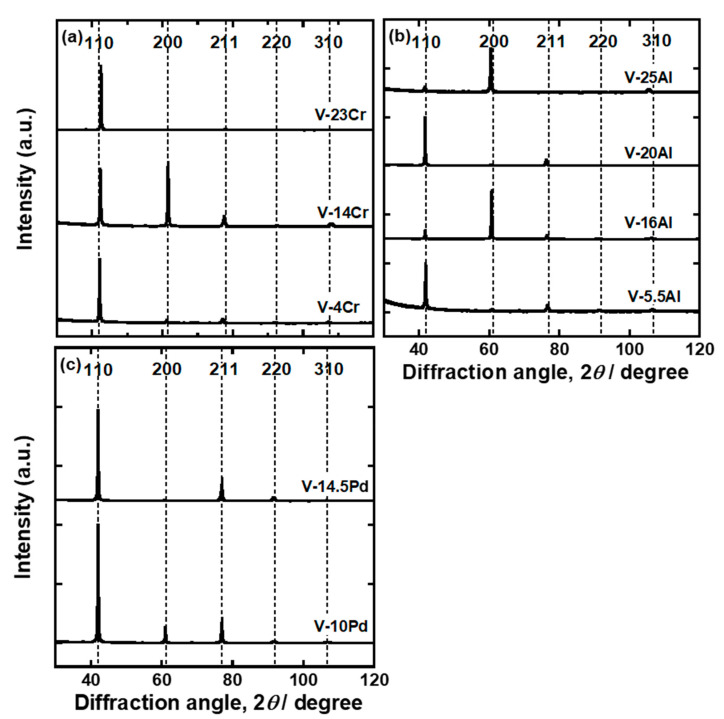
X-ray diffraction (XRD) profiles for (**a**) V-Cr, (**b**) V-Al, and (**c**) V-Pd alloys fabricated in this study.

**Figure 2 membranes-11-00067-f002:**
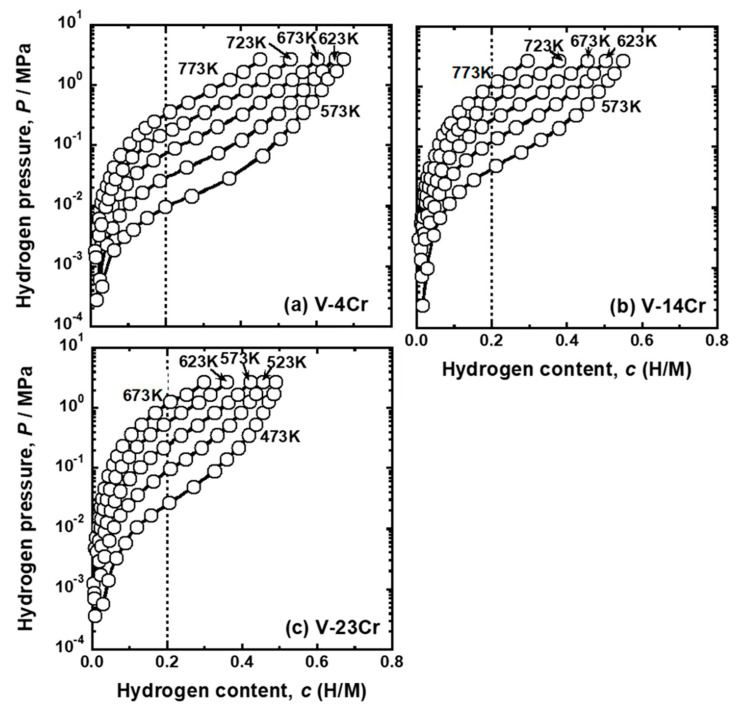
Pressure-composition isotherms (PCT curves) of (**a**) V-4mol%Cr, (**b**) V-14mol%Cr, and (**c**) V-23mol%Cr alloys.

**Figure 3 membranes-11-00067-f003:**
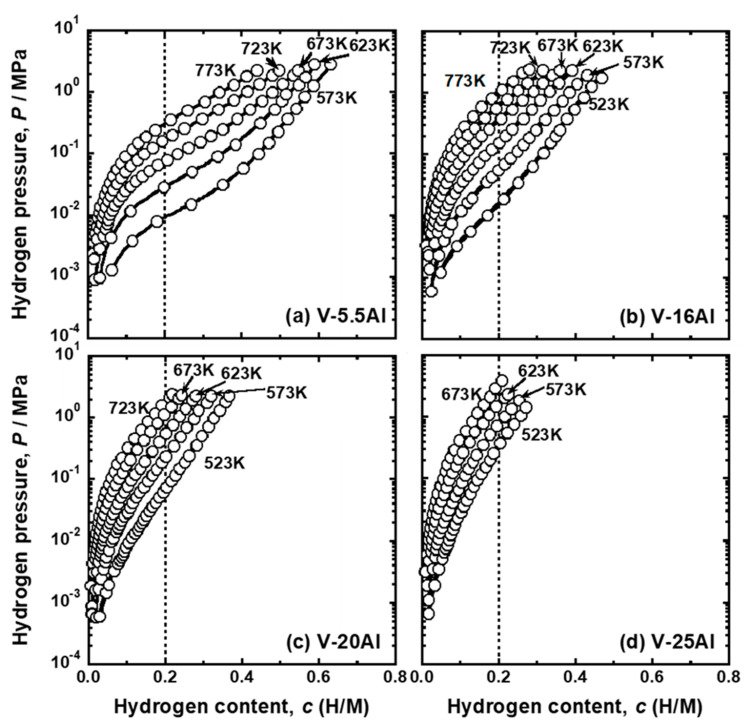
PCT curves of (**a**) V-5 mol%Al, (**b**) V-15 mol%Al, (**c**) V-20 mol%Al, and (**d**) V-25 mol%Al alloys.

**Figure 4 membranes-11-00067-f004:**
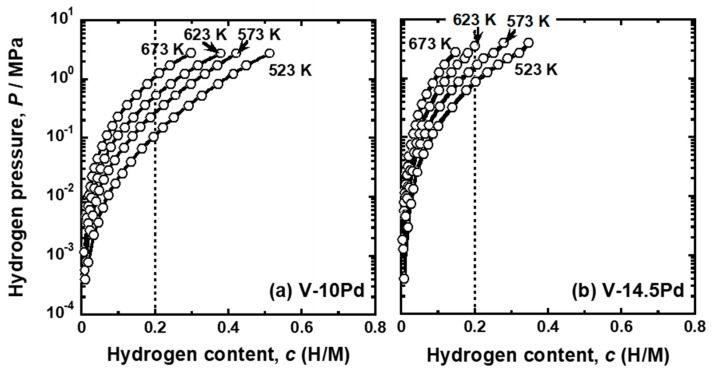
PCT curves of (**a**) V-10 mol%Pd and (**b**) V-14.5 mol%Pd alloys.

**Figure 5 membranes-11-00067-f005:**
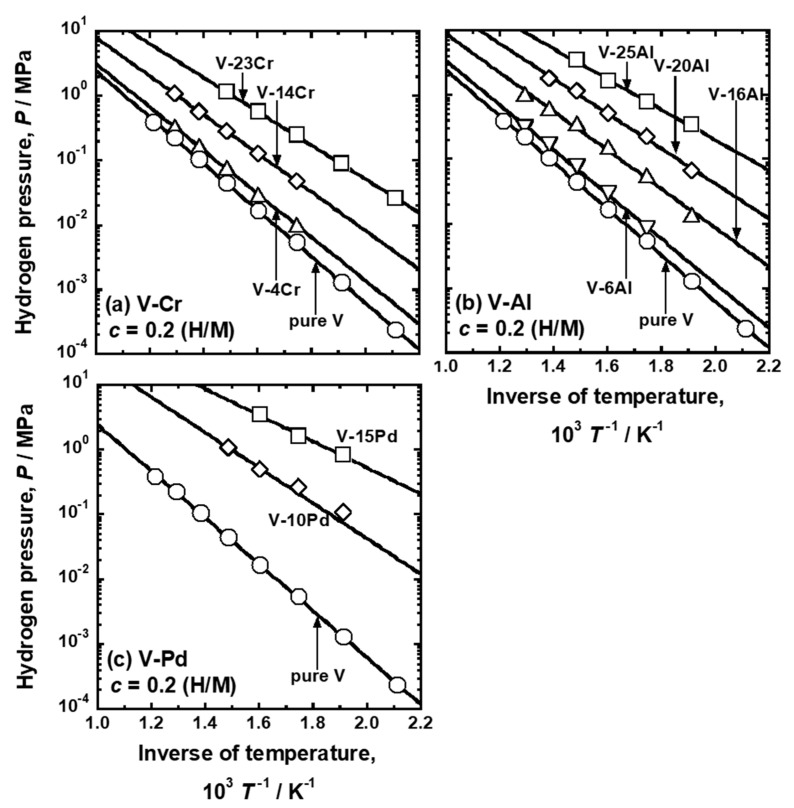
Relationship between the equilibrium hydrogen pressure (*P*) at 0.2 (H/M) and the inverse of temperature (*T*
^−1^) for (**a**) V-Al, (**b**) V-Cr, and (**c**) V-Pd alloys. The values for pure V [15] are also shown in the figures.

**Figure 6 membranes-11-00067-f006:**
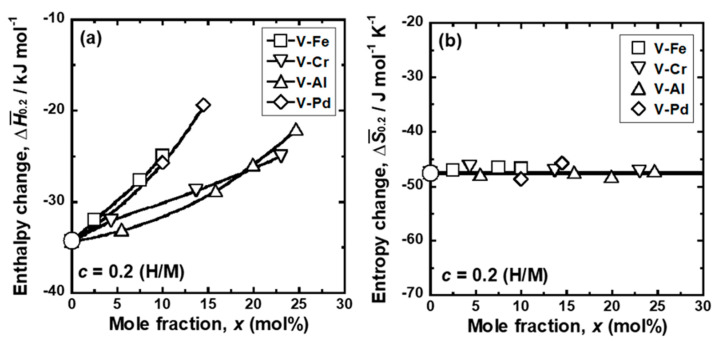
Changes in the (**a**) partial molar enthalpy change ($\Delta \bar{H}$
_0.2_) and (**b**) partial molar entropy change ($\Delta \bar{H}$_0.2_) for hydrogen dissolution at 0.2(H/M), as a function of the mole fraction (*x*) of alloying elements. For comparison, the results of pure V and V-Fe alloys are also shown in the figure [15].

**Figure 7 membranes-11-00067-f007:**
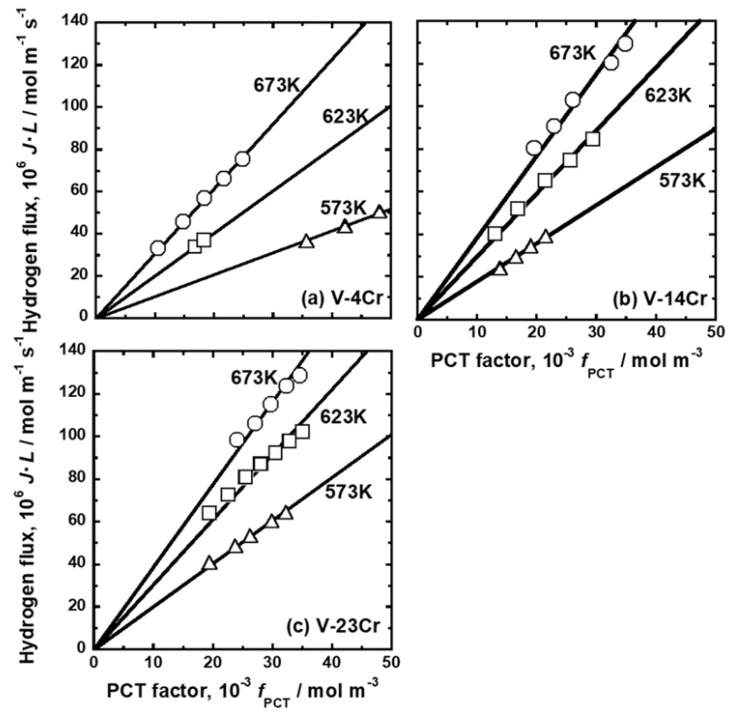
Relationship between the normalized hydrogen flux (*J·L*) and the PCT factor (*f*_PCT_): (**a**) V-4 mol%Cr, (**b**) V-14 mol%Cr, and (**c**) V-23 mol%Cr.

**Figure 8 membranes-11-00067-f008:**
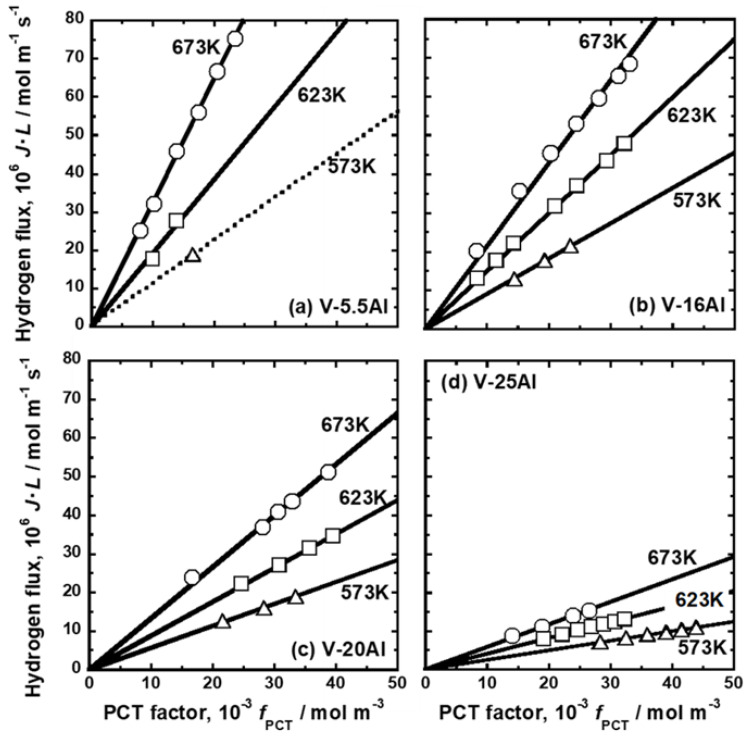
Relationship between the normalized hydrogen flux (*J·L*) and the PCT factor (*f*_PCT_): (**a**) V-5 mol%Al, (**b**) V-16 mol%Al, (**c**) V-20 mol%Al, and (**d**) V-25 mol%Al.

**Figure 9 membranes-11-00067-f009:**
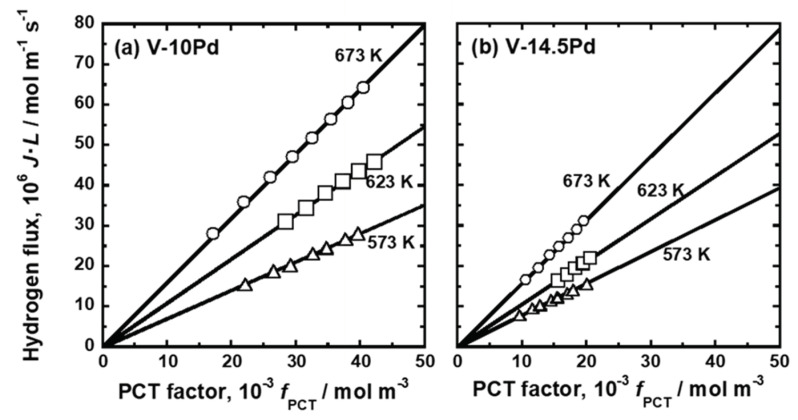
Relationship between the normalized hydrogen flux (*J·L*) and the PCT factor (*f*_PCT_): (**a**) V-10 mol%Pd and (**b**) V-15 mol%Pd.

**Figure 10 membranes-11-00067-f010:**
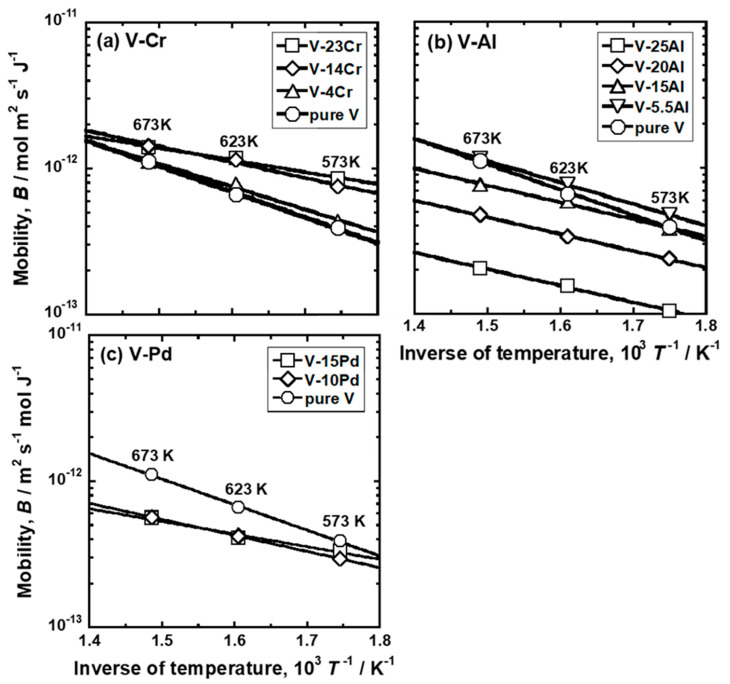
Arrhenius plots of the mobility for hydrogen diffusion in (**a**) V-Cr, (**b**) V-Al, and (**c**) V-Pd alloys. For comparison, the results of pure V are also shown in the figure [15].

**Figure 11 membranes-11-00067-f011:**
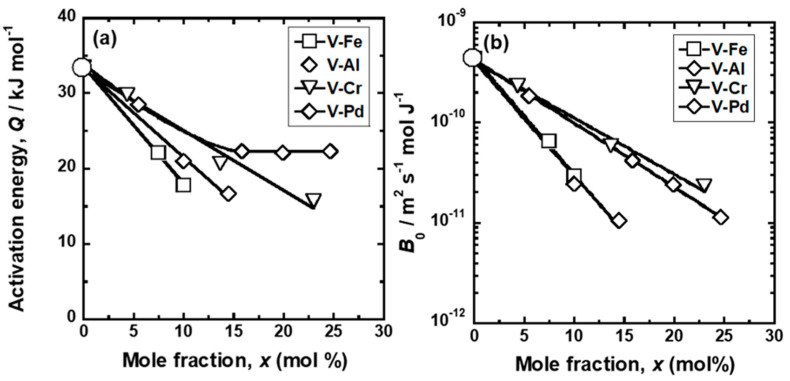
Changes in the (**a**) activation energy (*Q*) and (**b**) pre-exponential factor (*B*_0_) for hydrogen diffusion as a function of the mole fraction (*x*) of alloying elements. The values for pure V and V-Fe [15] are also shown in the figure.

**Figure 12 membranes-11-00067-f012:**
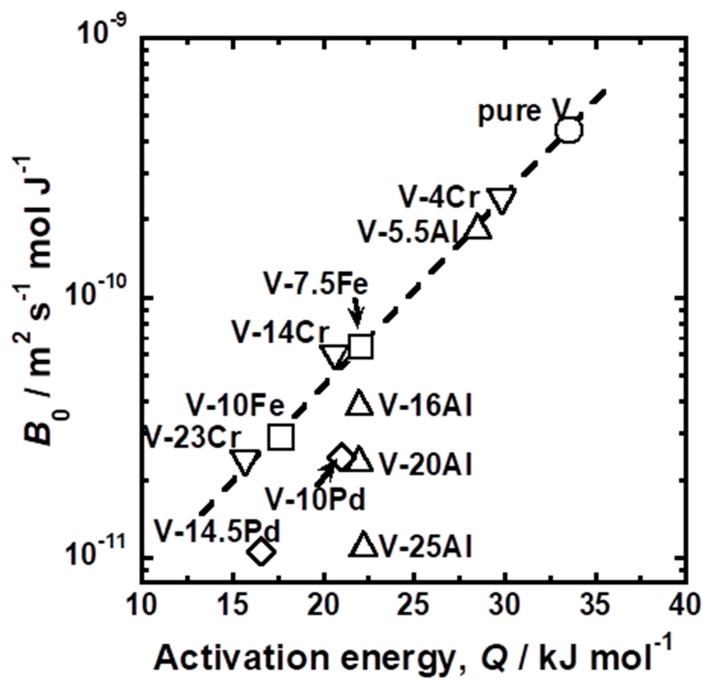
Relationship between the pre-exponential factor (*B*_0_) and the activation energy for hydrogen diffusion (*E*) in pure V and various V-based alloys.

**Figure 13 membranes-11-00067-f013:**
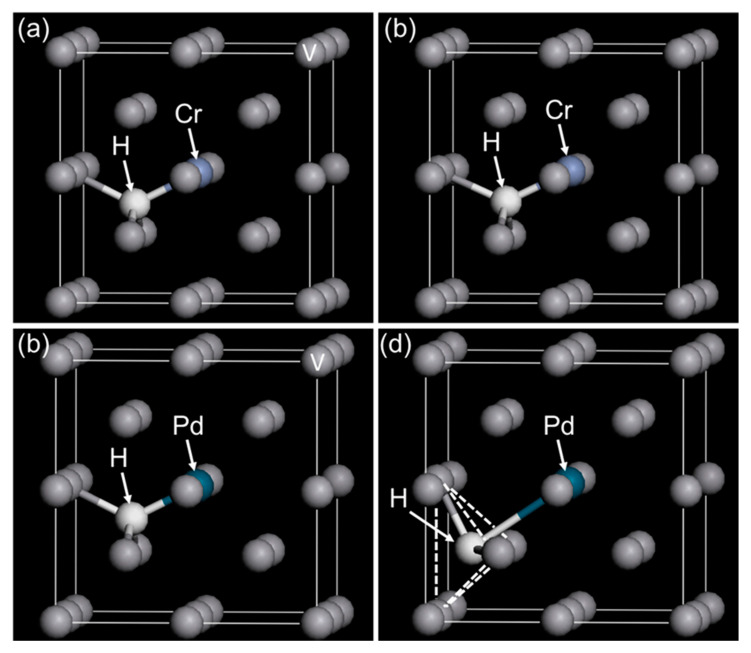
(**a**,**b**) V_15_Cr_1_H_1_ (T1) and (**c**,**d**) V_15_Pd_1_H_1_ models (T1) (**a**,**c**) before and (**b**,**d**) after geometry optimizations.

**Figure 14 membranes-11-00067-f014:**
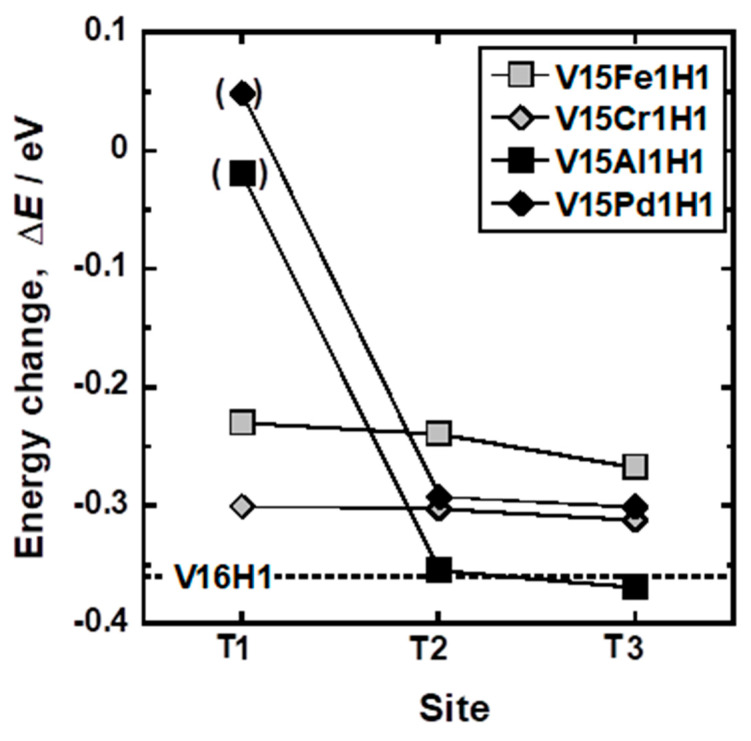
Total energy change (∆*E*) for insertion of a hydrogen atom into the T1, T2, and T3 sites of alloying elements (Fe, Al, Cr, and Pd).

**Figure 15 membranes-11-00067-f015:**
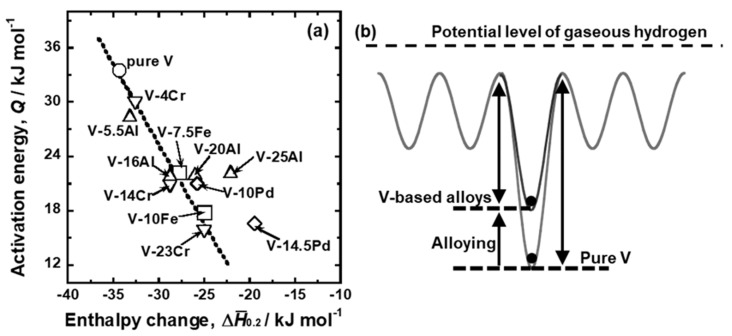
(**a**) Relationship between the activation energy for hydrogen diffusion (*Q*) and the partial molar enthalpy change of hydrogen for forming hydrogen solid solution with 0.2 (H/M) ($\Delta \bar{H}$
_0.2_) and (**b**) schematic illustration showing the potential level of a hydrogen atom at an interstitial T site in pure V and V-based alloys.

**Figure 16 membranes-11-00067-f016:**
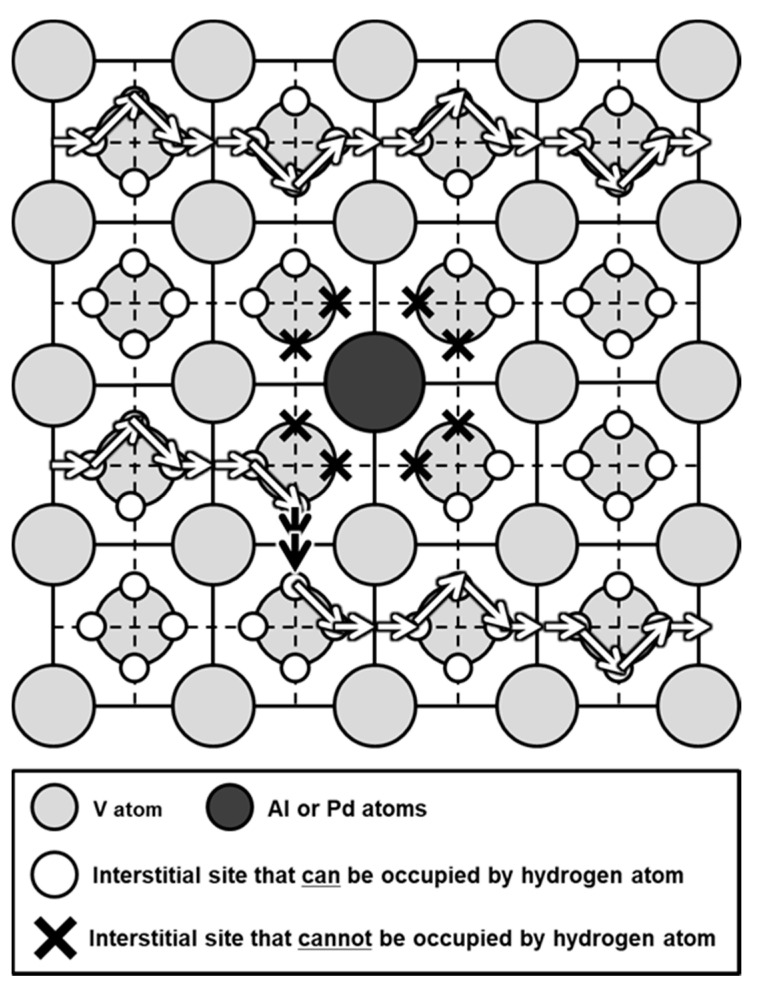
Schematic illustrations of (001) plane projection on a body-centered cubic (BCC) crystal lattice. Arrows represent the jumps of hydrogen atoms between adjacent interstitial sites.

**Table 1 membranes-11-00067-t001:** Chemical compositions of V-Cr, V-Al, and V-Pd alloys analyzed by SEM/EDS analysis.

Sample	Mole Fraction of Alloying Elements (mol%)
V-4Cr	4.3
V-14Cr	13.7
V-23Cr	23.0
V-5.5Al	5.5
V-16Al	15.8
V-20Al	19.9
V-25Al	24.7
V-10Pd	9.8
V-14.5Pd	14.5

**Table 2 membranes-11-00067-t002:** Temperature and pressure conditions of hydrogen permeation tests for V-Cr alloy membranes.

Sample	Temperature, *T*/K	Hydrogen Pressure, *P*/kPa	
		Feed	Permeate
V-4 Cr (*L* = 0.500 mm)	573	30, 35, 40	10
	623	28, 30	
	673	30, 40, 50, 60, 70	
V-14 Cr (*L* = 0.428 mm)	573	30, 35, 40, 45	
	623	45, 60, 80, 100, 120	
	673	120, 150, 180, 250, 280	
V-23 Cr (*L* = 0.521 mm)	573	100, 130, 150, 180, 200	
	623	200, 250, 300, 350, 400, 450, 500	
	673	500, 600, 700, 800, 900	

**Table 3 membranes-11-00067-t003:** Temperature and pressure conditions of hydrogen permeation tests for V-Al alloy membranes.

Sample	Temperature, *T*/K	Hydrogen Pressure, *P*/kPa	
		Feed	Permeate
V-5.5 Al (*L* = 0.550 mm)	573	20	10
	623	20, 25	
	673	25, 30, 40, 50, 60, 70	
V-16 Al (*L* = 0.562 mm)	573	30, 40, 50	
	623	30, 40, 50, 80, 100, 130, 150	
	673	50, 100, 150, 180, 200, 250, 300, 350	
V-20 Al (*L* = 0.425 mm)	573	100, 150, 200	
	623	200, 300, 400, 500	
	673	200, 500, 600, 700, 1000	
V-25 Al (*L* = 0.504 mm)	573	300, 400, 500, 600, 700, 800	
	623	200, 300, 400, 500, 600, 700, 800, 900	
	673	300, 500, 800, 1000	

**Table 4 membranes-11-00067-t004:** Temperature and pressure conditions of hydrogen permeation tests for V-Pd alloy membranes.

Sample	Temperature, *T*/K	Hydrogen Pressure, *P*/kPa	
		Feed	Permeate
V-10 Pd (*L* = 0.521 mm)	573	100, 130, 150, 180, 200, 230, 250	10
	623	250, 300, 350, 400, 450, 500	
	673	200, 300, 400, 500, 600, 700, 800, 900	
V-14.5 Pd (*L* = 0.366 mm)	573	100, 130, 150, 180, 200, 230, 250, 300	
	623	300, 350, 400, 450, 500	
	673	300, 400, 500, 600, 700, 800, 900	

**Table 5 membranes-11-00067-t005:** Mobility of hydrogen atoms quantified from the slope of the regression lines shown in Figure 7, Figure 8 and Figure 9.

Sample	Temperature, *T*/K	Mobility, 10^13^ *B*/mol m^2^ s^−1^ J^−1^
V-4 Cr	673	11.0
	623	7.8
	573	4.4
V-14 Cr	673	14.3
	623	11.4
	573	7.5
V-23 Cr	673	13.8
	623	11.7
	573	8.5
V-5.5 Al	673	11.5
	623	7.7
	573	4.8
V-16 Al	673	7.6
	623	5.8
	573	3.8
V-20 Al	673	4.7
	623	3.4
	573	2.4
V-25 Al	673	2.0
	623	1.6
	573	1.0
V-10 Pd	673	5.7
	623	4.2
	573	3.0
V-14.5 Pd	673	5.6
	623	4.1
	573	3.3
Pure Pd [28]	673	10.0
	573	6.0
Pd-23 Ag [28]	673	8.3
	623	6.4
	573	4.5

**Table 6 membranes-11-00067-t006:** Mobility of hydrogen atoms quantified from the slope of the regression lines shown in Figure 10.

Sample	Temperature Range	Pre-Exponential Factor, 10^11^ *B*_0_/mol m^2^ s^−1^ J^−1^	Activation Energy, *E*/kJ mol^−1^
V-4 Cr	573~673 K	23.3	29.8
V-14 Cr		5.9	20.6
V-23 Cr		2.3	15.6
V-5.5 Al	573~673 K	18.9	28.5
V-16 Al		4.0	22.0
V-20 Al		2.4	22.0
V-25 Al		1.1	22.2
V-10 Pd	573~673 K	2.4	21.0
V-14.5 Pd		1.1	16.6
Pure Pd [28]	423~673 K	1.4	14.9
Pd-23 Ag [28]	373~673 K	7.3	24.4

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
