# Peer review of "Quantitative Evaluations of Hydrogen Diffusivity in V-X (X = Cr, Al, Pd) Alloy Membranes Based on Hydrogen Chemical Potential"

_membranes, 2021, doi:10.3390/membranes11010067_

Round 1

Reviewer 1 Report

Typos that need to be addressed:

  1. line 318, 327, 378, 420: "energyrequired" needs to be changed to "energy required"
  2. line 108: "elminate" needs to be changed to "eliminate"
  3. line 365: "blockingrequired" needs to be changed to "blocking required"

Comments:

  1. What is the predicted selectivity over other small gases? The utilization of these membranes is strongly depended on selectivity rather than high hydrogen flux. 
  2. Are these membranes expected to have resistance over impurities such as CO and H2S? Common applications of these membranes would be in the gas industry where there are gas streams that contain high percentages of the aforementioned impurities. 

Author Response

We really appreciate the reviewer's comments and suggestions which helped us to improve the manuscript. We revised our manuscript based on the comments and suggestions. The revised parts are highlighted in yellow. The detailed answers to the comments are listed below.

Typos

line 318, 327, 378, 420: "energyrequired" needs to be changed to "energy required"

line 108: "elminate" needs to be changed to "eliminate"

line 365: "blockingrequired" needs to be changed to "blocking required"

Answer:

We really appreciate the reviewer’s careful check for our manuscript. We modified the pointed typos (highlighted in yellow). 

Comments 1:

What is the predicted selectivity over other small gases? The utilization of these membranes is strongly depended on selectivity rather than high hydrogen flux.

Answer:

In the porous membranes with fine pores to permeate hydrogen, selectivity is very important as the reviewer pointed out. However, in the case of the dense metallic membranes, the permeability of gases other than hydrogen is extremely low because of their low solubility and diffusivity in the metallic membranes. For example, while hydrogen atoms diffuse through the vanadium membrane with 100 um thickness, the carbon element can diffuse through only one or two vanadium bcc crystal lattice.  Therefore, the hydrogen selectivity is almost infinite theoretically, and hydrogen permeability is important for the utilization.  

Comments 2:

Are these membranes expected to have resistance over impurities such as CO and H2S? Common applications of these membranes would be in the gas industry where there are gas streams that contain high percentages of the aforementioned impurities.

Answer:

It is an important comment. As the reviewer pointed out the effect of impurity gases on the hydrogen permeability is directly important for the applications of the membranes. In the case of non-Pd-based alloy membranes, among which V-based alloy membranes were investigated in this study, the catalytic overlayers are coated on the surface of the membranes. Therefore, the effect of impurity gases is dominated by the interactions between the overlayers and impurity gases. In this study, Pd-27 mol%Ag alloy was coated on the V-based alloy membranes. The Pd-27 mol%Ag alloy exhibit high resistance to CO and H2O gas around the measurement temperature in this study [R1]. However, Pd-27 mol%Ag alloy is strongly affected by the presence of H2S gas because of the formation of sulfides [R2]. To mitigate the effect of H2S gas, Pd-Cu alloy overlayers are effective [R3]. Now, we investigate the relationship between kinds of overlayers and stability of hydrogen permeability and will publish in the future.

[R1]Nguyen, T.H.; Mori, S.; Suzuki, M. Hydrogen permeance and the effect of H2O and CO on the permeability of Pd0.75Ag0.25 membranes under gas-driven permeation and plasma-driven permeation. Chem. Eng. J. 2009, 155, 55–61.

[R2]Peters, T.A.; Stange, M.; Veenstra, P.; Nijmeijer, A.; Bredesen, R. The performance of Pd-Ag alloy membranes films under exposure to trace amount of H2S. 2016, 499, 105-115.

[R3]Yang, J.Y.; Nishimura C.; Komaki, M. Hydrogen permeation of Pd60Cu40 alloy covered V-15Ni composite membrane in mixed gases containing H2S. 2008, 309, 246-250. 

Asuka Suzuki

9 January 2020

Reviewer 2 Report

Manuscript Number: membranes-1070753

Title: Quantitative evaluations of hydrogen diffusivity in V-X (X = Cr, Al, Pd) alloy membranes based on hydrogen chemical potential

This manuscript reports a study of the hydrogen diffusivity in V-based alloy membranes using hydrogen chemical potential. To accept this paper, the following issues should be addressed:

1) Figure 1: I would like to suggest moving Figure 1 from Section 2 to Section 3.

2) Table 2: is the "Hydrogen Pressure" actually the partial pressure of hydrogen? And it should be "Permeate" rather than "Permeation".

3) Section 2: why the hydrogen pressure increases with the increasing mole fraction of alloying elements?

4) Section 3: please correct the sub-title: "2.1" and "2.2".

5) It is strongly recommended to add a comparison between the H2 diffusivity obtained in this study with H2 diffusivity in Pd membranes, either in a table or in the main text.

Author Response

We really appreciate the reviewer's variable comments and suggestions which helped us to improve the manuscript. We revised our manuscript based on the comments and suggestions. The revised parts are highlighted in yellow. The detailed answers to the comments are listed below.

Comments 1:

Figure 1: I would like to suggest moving Figure 1 from Section 2 to Section 3.

Answer:

We appreciate kind suggestion of the reviewer. We moved Figure 1 to Section 3.

Comments 2:

2) Table 2: is the "Hydrogen Pressure" actually the partial pressure of hydrogen? And it should be "Permeate" rather than "Permeation".

Answer:

For the measurements of this study, we use ultra-high purity (G1 grade) hydrogen gas. Therefore, the total pressure is almost the same as the partial pressure of hydrogen. Therefore, we use the words “Hydrogen pressure”. Instead, sentences that we used G1 grade hydrogen gas were added in the section 2. The “Permeation” was modified into “Permeate”.

Comments 3:

Section 2: why the hydrogen pressure increases with the increasing mole fraction of alloying elements?

Answer:

The alloying elements in the present study have lower affinity to hydrogen than vanadium and destabilize the hydrogen atoms in interstitial sites in vanadium. So, the chemical potential of hydrogen atoms in vanadium increases with increasing the mole fraction of alloying elements. In the PCT measurements, the chemical potentials of the hydrogen atoms in metals and gaseous hydrogen are equal due to the equilibrium condition. Therefore, the increase in the chemical potential of hydrogen atoms in vanadium also increases the chemical potential of gaseous hydrogen in equilibrium. Since the chemical potential of gaseous hydrogen has a positive correlation with the hydrogen pressure, the pressure of hydrogen increases.

Comments 4:

Section 3: please correct the sub-title: "2.1" and "2.2".

Answer:

We really appreciate the reviewer’s careful check. We modified “2.1” and “2.2” into “3.1” and “3.2”.

Comments 5:

It is strongly recommended to add a comparison between the H2 diffusivity obtained in this study with H2 diffusivity in Pd membranes, either in a table or in the main text.

Answer:

Thanks to the reviewer for important recommendation. We added the mobility, activation energy, and pre-exponential factor of pure Pd and Pd-23 mol%Ag in Table 5 and 6. We also added the sentences in the main text.

Asuka Suzuki

9 January 2020